# Impact of Small Molecules on Intermolecular G-Quadruplex Formation

**DOI:** 10.3390/molecules24081570

**Published:** 2019-04-20

**Authors:** Prabesh Gyawali, Keshav GC, Yue Ma, Sanjaya Abeysirigunawardena, Kazuo Nagasawa, Hamza Balci

**Affiliations:** 1Department of Physics, Kent State University, Kent, OH 44242, USA; pgyawal2@kent.edu; 2Department of Chemistry and Biochemistry, Kent State University, Kent, OH 44242, USA; kgc@kent.edu (K.G.); sabeysir@kent.edu (S.A.); 3Department of Biotechnology and Life Science, Tokyo University of Agriculture and Technology, Koganei, Tokyo 184-8588, Japan; yue-ma@m2.tuat.ac.jp (Y.M.); knaga@cc.tuat.ac.jp (K.N.)

**Keywords:** G-quadruplex, small molecule, single molecule, FRET, biosensor

## Abstract

We performed single molecule studies to investigate the impact of several prominent small molecules (the oxazole telomestatin derivative L2H2-6OTD, pyridostatin, and Phen-DC_3_) on intermolecular G-quadruplex (i-GQ) formation between two guanine-rich DNA strands that had 3-GGG repeats in one strand and 1-GGG repeat in the other (3+1 GGG), or 2-GGG repeats in each strand (2+2 GGG). Such structures are not only physiologically significant but have recently found use in various biotechnology applications, ranging from DNA-based wires to chemical sensors. Understanding the extent of stability imparted by small molecules on i-GQ structures, has implications for these applications. The small molecules resulted in different levels of enhancement in i-GQ formation, depending on the small molecule and arrangement of GGG repeats. The largest enhancement we observed was in the 3+1 GGG arrangement, where i-GQ formation increased by an order of magnitude, in the presence of L2H2-6OTD. On the other hand, the enhancement was limited to three-fold with Pyridostatin (PDS) or less for the other small molecules in the 2+2 GGG repeat case. By demonstrating detection of i-GQ formation at the single molecule level, our studies illustrate the feasibility to develop more sensitive sensors that could operate with limited quantities of materials.

## 1. Introduction

G-quadruplex structures (GQs) have emerged not only as promising targets for cancer therapy [1,2] or transcription-level [3,4,5] and translation-level [6] gene expression regulation, but they have also found use in various biotechnology applications [7]. As physiological relevance and technological potential of GQs became better established, research on identifying and synthesizing small molecules (SMs) that stabilize them, also experienced significant progress [8,9,10,11]. Prominence of GQ stabilizing SMs is typically characterized by their specificity to GQs, compared to double-stranded or single-stranded DNA (dsDNA or ssDNA), and by the increase these give rise to, in the thermal melting point (ΔT_m_) of GQs. Pyridostatin (PDS), Phen-DC3, and oxazole telomestatin derivatives (OTD) are some of the prominent SMs in these respects (Figure 1A shows chemical structures of these SMs) [12,13,14,15]. In the particular context of human telomeric GQ (hGQ), ΔT_m_ = 30 °C for Phen-DC_3_ [16], ΔT_m_ = 16 °C for L2H2-6OTD [17], and ΔT_m_ = 35 °C for PDS [18], have been reported under different ionic conditions (50–150 mM KCl).

Even though SMs are conventionally designed to stabilize intramolecular GQs, they may also facilitate intermolecular GQ (i-GQ) formation between G-rich neighboring regions of complementary strands. Bioinformatics studies showed that potential GQ-forming sequences (PQS), in which both strands of genomic DNA are involved in the GQ formation, co-localize with the functional sites in human genome and are more abundant than PQS within a single DNA strand [19]. The SMs that are designed to stabilize intramolecular GQs, could also promote and stabilize i-GQ between such strands, which may or may not be a desired effect, depending on the application. In addition, the concept of i-GQ formation by two or more strands of DNA has been used in various biotechnological applications, including G-wires [20,21], DNA-origami structures [22,23], and DNA-based devices, such as DNA walkers [24] and Heme-DNAzymes [25,26]. Biosensors that incorporate i-GQ structures, have been used to detect the presence of p53 or BRCA1 gene fragment [27,28], or deletion of the *LMP1* gene fragment [29,30], a metal ion [31], or a chemical [32,33]. In such applications, the two G-rich strands are brought to close-enough proximity to form a “split G-quadruplex” if one of the conditions listed above is satisfied. Whether or not an i-GQ is formed, can be detected via (i) a probe, such as an iridium compound that demonstrates enhanced luminescence upon binding to i-GQ [32]; (ii) the chemiluminescence signal that results from activation of the DNAzyme formed by i-GQ and hemin, in the presence of a metal ion or a chemical [31]; and (iii) a change in fluorescence signal, such as the quenching of fluorophores that are brought in close proximity upon i-GQ formation [28,34]. Since the performance of such a sensor depends on the efficiency of i-GQ formation, optimizing the sequences of the strands or providing SMs that facilitate i-GQ formation, would result in more sensitive sensors. There has been significant efforts for identifying and synthesizing GQ-stabilizing SMs [11,14,35,36,37], and improvements in their functionality, for different applications [38,39,40,41] where intramolecular GQs have generally been used as the scaffold. Whether such compounds function as efficiently in stabilizing i-GQ, is one of the questions we investigated in this study. Our particular focus was on detection, using single molecule methods, which provide unique capabilities and challenges, compared to ensemble level measurements. In particular, the enhanced sensitivity and low demand on materials are two promising aspects of using a single molecule scheme in sensor applications that involve i-GQ formation.

## 2. Results and Discussion

In order to study the impact of SMs on the i-GQ formation, we developed an smFRET assay where i-GQ formation is studied in the absence or presence of SMs. The DNA constructs and labeling positions were selected such that, formation of i-GQ between a surface-immobilized partial duplex DNA (pd-DNA) and a strand that is introduced into the chamber, results in a detectable increase in the FRET efficiency (E_FRET_). We studied two different configurations that involved telomeric repeats. In one case, three GGG repeats were in the surface-bound DNA (pd-S3) and another GGG-repeat was introduced as an additional strand (S1). This is called the 3+1 GGG configuration. In the other case, the surface-bound DNA (pd-S2) had two GGG-repeats and two additional GGG-repeats were introduced in another strand (S2). This is called the 2+2 GGG configuration. The 1+3 GGG configuration (monitoring FRET shift across pd-S1 while S3 is titrated) was not studied, since FRET efficiency would approach saturation for pd-S1, even due to the very short overhang (<10 nt), even before adding S3. Additionally, we did not explore any configuration that had more than three GGG repeats in one strand as that would result in the formation of an intramolecular GQ, which was not the focus of this study.

### 2.1. Influence of SMs on the i-GQ Formation in 3+1 GGG-Repeat Configuration

Figure 1 shows smFRET measurements where pd-S3 is immobilized to the surface and S1 is titrated at 150 mM K^+^. In the absence of S1, pd-S3 shows a FRET distribution that is peaked at E_FRET_ = 0.63 ± 0.05, the uncertainty being the sigma value of the Gaussian fit. Adding increasing concentration of S1, results in the emergence and gradual population of a second peak at E_FRET_ = 0.75 ± 0.06. The dashed red lines marked these peak positions. To illustrate the change in the FRET distribution more clearly, the 0 M S1 and 10 μM S1 data were overlaid in Figure 1C. We interpreted this newly emerging high FRET population as an evidence for the i-GQ formation, which resulted in the compaction of pd-S3, in the form of a G-triplex (GT). In order to ensure that the observed increase in the FRET efficiency was due to the i-GQ formation, similar measurements were performed in 150 mM LiCl or 150 mM TMAA, in which the i-GQ formation was not expected or was expected to be much weaker than that in the 150 mM K^+^. In both cases the FRET peak did not shift to higher values, even at 10 μM S1 (Figure 2A), suggesting that the compaction we observed in K^+^ was due to the i-GQ formation. To quantify how the fraction of this high FRET population changed with S1 concentration ([S1]), we subtracted the FRET distribution at 0 M S1, from the FRET distributions at the respective [S1]. As both FRET distributions were normalized to have a total area of 100%, the resulting subtraction histogram had a net zero area, with the negative areas representing depleted states and positive areas representing newly emerging states, upon adding S1. An example of the subtraction histogram is shown in the inset of Figure 1D. Integrating the green bins in this subtraction histogram, resulted in the total percentage of newly emerging states, which we interpreted as a measure of the i-GQ population. By plotting this positive area as a function of [S1] and performing a Hill function fit, an equilibrium constant could be obtained. Figure 1D shows the results of this analysis, with K_eq_ = 1.7 ± 0.1 μM. An alternative method to analyze this type of data was to fit the FRET histograms to a function with two Gaussian peaks and monitor how these populations varied with [S1]. However, the close proximity of the peaks, E_FRET_ = 0.63 ± 0.05 and E_FRET_ = 0.75 ± 0.06, made this approach prone to fitting errors.

In order to test whether the SMs had any effect on this type of i-GQ formation, we performed the same S1 titration in the presence of SM. We used an oxazole telomestatin derivative (L2H2-6OTD), PDS, and Phen-DC3 as model SMs, at 1 μM concentration. Among these SMs, only the L2H2-6OTD demonstrated significant improvement in the GQ formation and stability, while the other two SMs had unexpected effects on the smFRET distributions that prevented reliably quantifying their impact. The data on the L2H2-6OTD have been presented before the PDS and the Phen-DC_3_ have been discussed. Figure 1E shows S1 titration in the presence of 1 μM L2H2-6OTD, Figure 1F shows the overlay of 0 M and 10 μM S1 data, and Figure 1G shows the results of the subtraction analysis, where K_eq_ = 0.2 ± 0.1 μM was obtained. This K_eq_ is an order of magnitude smaller than that observed in the absence of SMs, suggesting a significant enhancement in the i-GQ formation in the presence of L2H2-6OTD. Similar measurements were also performed, using electrophoretic mobility shift assay (EMSA), where the S3 (GGGTTAGGGTTAGGG) strand was radiolabeled with phosphorus-32 (P32). Titrating S1 strand while keeping the S3 concentration at 10 nM, resulted in the emergence of a new band. These data are shown in Appendix A.

Unlike L2H2-6OTD, the impact of Phen-DC_3_ and PDS on the i-GQ formation in the 3+1 GGG configuration, could not be quantified. Upon adding these SMs (before adding S1), either a new low FRET peak or a broad distribution of low FRET states emerged (Figure 2B,C). The reason behind these low FRET states was not clear. Quenching of the acceptor fluorophore by the SM was one of the possibilities, as it was demonstrated that PDS and Phen-DC_3_ had such an effect when they bound to the close proximity of Cy5 [15]. As these low FRET states were observed before S1 was added, PDS and Phen-DC_3_ would have needed to interact with the intermediate folding states, such as G-triplex. Such low FRET states were not observed when the PDS was introduced in a chamber that contained a DNA construct with an unstructured overhang (pd-21T), of similar length, to that of the pd-S3 construct (Figure 2D).

### 2.2. Influence of SMs on the i-GQ Formation in the 2+2 GGG-Repeat Configuration

We also tested the impact of SMs on the i-GQ formation between pd-S2 and S2 (see Table 1 for sequences and Figure 3A for a schematic of the assay), using EMSA (Appendix A) and smFRET. At 150 mM K^+^, introducing higher concentrations of S2 resulted in the compaction of the structure in pd-S2, as demonstrated by a shift to higher FRET values in Figure 3B. In the absence of S2, pd-S2 demonstrated a peak at E_FRET_ = 0.78 ± 0.04. Introducing S2 resulted in the emergence and gradual population of a higher FRET peak at E_FRET_ = 0.87 ± 0.02. Figure 3C shows the histograms at 0 M and 10 μM S2, which demonstrated this shift. We interpreted this compaction to be due to the formation of the i-GQ between the pd-S2 and S2. Figure 3D shows the results of the subtraction analysis where the FRET distribution at 0 M S2 was subtracted from that at the respective [S2]. A Hill function fit resulted in K_eq_ = 2.3 ± 0.5 μM. We then repeated these measurements in the presence of 1 μM L2H2-6OTD, PDS or Phen-DC_3_, while all other ingredients of the imaging buffer were kept identical (Figure 3E–M). Figure 3F,I,L show the shift in the FRET histograms at 0 M and 10 μM S2 for L2H2-6OTD, PDS, and Phen-DC3, respectively. Similar subtraction analysis and Hill function fit the result in K_eq_ = 2.1 ± 0.2 μM for L2H2-6OTD (Figure 3G), K_eq_ = 0.8 ± 0.3 μM for PDS (Figure 3J), and K_eq_ = 2.2 ± 0.4 μM for Phen-DC3 (Figure 3M). Unlike the L2H2-6OTD and Phen-DC3, which did not result in a significant enhancement of the i-GQ formation in the 2+2 GGG configuration, the PDS resulted in a three-fold enhancement.

Overall, these results suggested that these SMs either did not facilitate the i-GQ formation in the 2+2 GGG configuration or had a relatively weaker impact, compared to the order of magnitude enhancement observed in the 3+1 GGG configuration. It is particularly interesting that L2H2-6OTD enhanced the i-GQ formation in the 3+1 GGG configuration, by an order of magnitude, but essentially had no impact on the i-GQ formation in the 2+2 GGG configuration. The reasons behind this very interesting and potentially significant observation was not clear, but there were several different possibilities. It was possible that the folding conformation attained in the 3+1 GGG configuration was more favorable to the L2H2-OTD stacking, compared to the conformation in the 2+2 GGG configuration. Another possibility was the higher stability of the i-GQ structure formed in the 3+1 GGG configuration, which resulted in smaller fluctuations in the structure and allowed stacking of the L2H2-6OTD. This might have been particularly important for the transiently folded i-GQ structures, which might have been stable enough to be bound and further stabilized by the L2H2-6OTD in the case of the 3+1 GGG but not in the case of 2+2 GGG. Finally, it is possible that L2H2-6OTD facilitated the transition to a fully folded i-GQ from certain intermediate folding states that were more likely to form in the 3+1 GGG configuration, compared to the 2+2 GGG configuration, such as the G-triplex. It is likely that different scenarios were prevalent for different SMs, based on their structural details, where on the i-GQ structure, they bound their affinity to the intermediate folding states and their tolerance to the fluctuations in the structure. A conclusive answer to this interesting question would require both structural studies and dynamic studies with a high enough time resolution to probe the i-GQ folding process.

## 3. Materials and Methods

Prism-type total internal reflection fluorescence microscopy measurements were performed on an Olympus IX-71 microscope, equipped with an Olympus 60x, 1.20 NA water objective, and an Andor Ixon EMCCD camera (iXon DV 887-BI EMCCD, Andor Technology, South Windsor, CT, USA). A λ = 532 nm green laser (SpectraPhysics Excelsior) was used as the excitation source. A total of 1% of PEG molecules were tagged with biotin to provide attachment points for the biotinylated DNA molecules, which bind to the biotin-PEG via neutravidin. To provide adequate statistics for the analysis, 250–350 DNA molecules per imaging area (~5 × 10^3^ µm^2^) were targeted as the ideal density in most measurements. The imaging buffer contained Tris base (50 mM, pH 7.5), 2 mM Trolox, 0.8 mg/mL glucose, 0.1 mg/mL bovine serum albumin (BSA), 0.1 mg/mL glucose oxidase, 0.02 mg/mL catalase, 150 mM KCl and 2 mM MgCl_2_. LiCl or TMAA [42] were used, instead of KCl in some control measurements. A total of 1 μM SM and varying concentrations of the second DNA strand (S1 or S2) were also added to the imaging buffer, in relevant measurements. Recording of the data started after incubating all components in the chamber, for 15 minutes. Short movies (15 frames) or long movies (1500–3000 frames) were recorded at a frame integration time of 100 msec. Each of the histograms in Figure 1, Figure 2 and Figure 3 was constructed from single molecule data, obtained from 20–30 different movies, each imaging a different site on the surface, and represented the distribution of the FRET efficiency for several thousand molecules.

Both, the fluorescently-labeled and the unlabeled DNA oligonucleotides were purchased as PAGE or HPLC, purified from the Integrated DNA Technologies (IDT). In order to make the pdDNA constructs, the Stem strand (Table 1) was annealed with the relevant strand at 90 °C, for 3 min, followed by a slow cooling to room temperature, over 2–3 h.

## 4. Conclusions

This study demonstrated the feasibility, sensitivity, and material demands of detecting the i-GQ formation, using a single molecule platform in the presence or absence of GQ stabilizing SMs. It presented the range of impact that could be expected when SMs were utilized to enhance the i-GQ formation for different configurations of GGG repeats. An order of magnitude enhancement in the i-GQ formation was observed in 3+1 GGG configuration with the L2H2-6OTD. The 2+2 GGG configuration was less efficient at the i-GQ formation, with or without SMs. The largest enhancement in this configuration was three-fold with PDS. This was reasonable, as it would be easier for intermediate folding conformations (predecessors of the i-GQ) to form, when three GGG repeats were in one strand. In a physiological setting, these results could be desired for some applications, e.g., when elevated synthetic lethality and genomic instability in cancer cells was desired. However, in applications where specific targeting of a particular GQ was desired, e.g., to regulate transcription level gene expression, then these SMs might have given rise to the i-GQ formation at sites where they would otherwise not form. Our study also had consequences for biotechnology applications that utilized and depended on the i-GQ formation. We demonstrated that with the proper design of DNA constructs and the right SM, it was possible to enhance the i-GQ formation, hence, the signal of a sensor could also be enhanced by an order of magnitude. Finally, fluorescence-based detection schemes needed to take into account the potential photophysical effects that might arise due to SM-dye and dye-GQ interactions [43]. While the L2H2-6OTD did not have these limitations, both Phen-DC3 and PDS gave rise to complications. Therefore, if these molecules were to be used for sensor applications, either possible quenching of the fluorophores should be well-characterized or non-fluorescence-based methods should be used for detection.

## Figures and Tables

**Figure 1 molecules-24-01570-f001:**
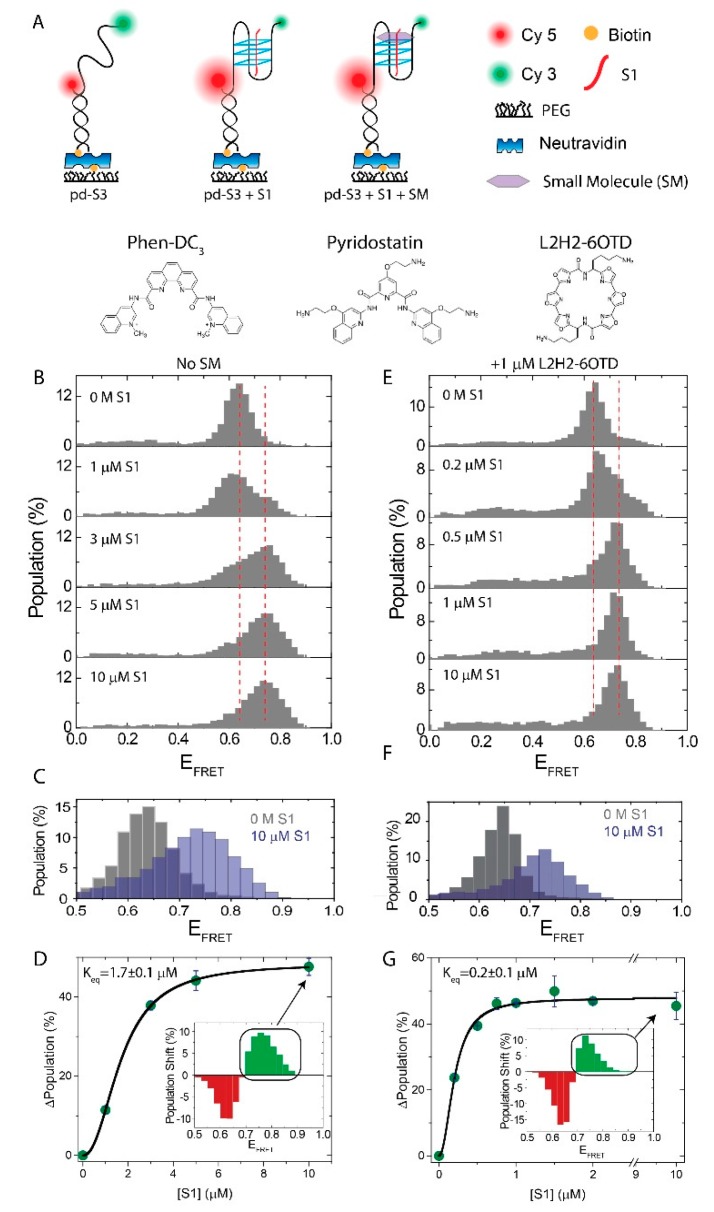
The effect of L2H2-6OTD on the i-GQ formation in 3+1 GGG configuration in the 150 mM K^+^. (**A**) Schematic of the DNA constructs and the smFRET assay. The bottom panel shows the chemical structures of the SMs used in the measurements. (**B**) S1 was titrated from 0 M to 10 μM in the absence of SM. Higher FRET states were populated as [S1] increased, suggesting i-GQ formation. The red dashed lines indicate the peak of the FRET distribution at 0 M and 10 μM S1. (**C**) 0 M and 10 μM S1 data were overlaid to illustrate the shift. (**D**) The transition to the i-GQ state was quantified by subtracting the FRET distribution at 0 M S1 from that at corresponding [S1]. The total positive higher FRET population (green bars at the inset) was plotted as a function of [S1]. The black line was a Hill function fit. (**E**–**G**) were the equivalents of (**B**–**D**), respectively, when 1 μM L2H2-6OTD was maintained in the chamber, during S1 titration. K_eq_ decreased from 1.7 μM in the absence of SM to 0.2 μM in the presence of 1 μM L2H2-6OTD.

**Figure 2 molecules-24-01570-f002:**
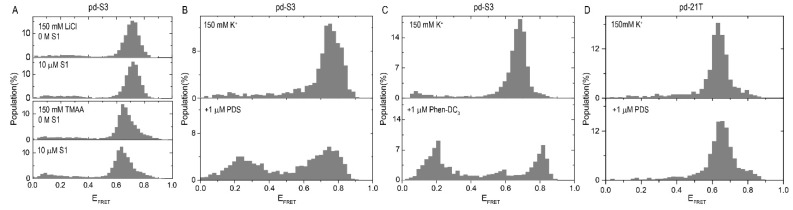
(**A**) Even in the presence of 10 μM S1, no shift in the FRET was observed for the pd-S3 construct in the 150 mM LiCl (top) or 150 mM TMAA (bottom), suggesting the shift in KCl to be due to the i-GQ formation. (**B**) Demonstration of the change in the FRET histogram upon adding 1 μM PDS. A significant low FRET population emerged which was considered to be due to the interaction of PDS with the fluorophores. (**C**) Similar low FRET states were observed when 1 μM Phen-DC_3_ was introduced. (**D**) Such low FRET states were not observed when PDS was introduced in a chamber that contained pd-21T construct, an unstructured DNA (polythymidine) of similar length to pd-S3.

**Figure 3 molecules-24-01570-f003:**
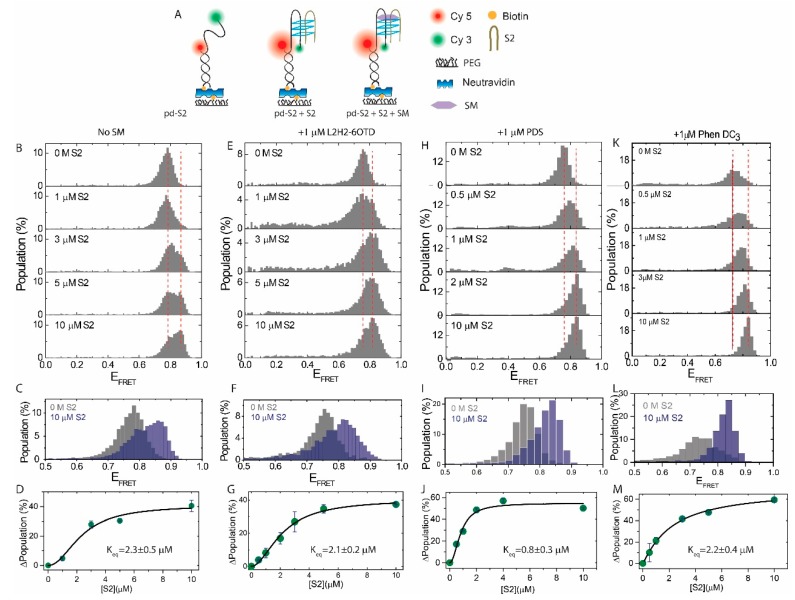
The effect of SMs on i-GQ formation in 2+2 GGG-repeat configuration in 150 mM K^+^. (**A**) Schematic of the DNA constructs and the smFRET assay. (**B**) S2 was titrated from 0 M to 10 μM in the absence of SM. Higher FRET states were populated in the FRET histogram as [S2] increased, suggesting i-GQ formation. The two red dashed lines indicated the peak of the FRET distribution at 0 M and 10 μM S2. (**C**) 0 M and 10 μM S2 data were overlaid to illustrate the shift. (**D**) Formation of i-GQ state was quantified by subtracting the FRET distribution at 0 M S2, from that at the corresponding [S2]. The cumulative of positive higher FRET population was plotted as a function of [S2]. The black line was a Hill function fit. (**E**–**G**) were identical to (**B**–**D**), respectively, except 1 μM L2H2-6OTD was maintained in the chamber during S2 titration. (**H**–**J**) showed the same measurements in the presence of 1 μM PDS, and (**K**–**M**) in the presence of 1 μM Phen-DC_3_.

**Table 1 molecules-24-01570-t001:** DNA sequences used for this study. The sequences that formed the stem are shown in bold fonts. Stem is complementary to an 18nt segment in the pd-S3, pd-S2, and pd-21T.

Construct	Sequence (5′–3′)
Stem	Cy5-*GCCTCGCTGCCGTCGCCA*-Biotin
pd-S3	*TGGCGACGGCAGCGAGG* AGGGTTAGGGTTAGGGTTAG-Cy3 + Stem
pd-S2	*TGG CGA CGG CAG CGA GGC* AGGGTTAGGGTTAG-Cy3 + Stem
pd-21T	*TGG CGA CGG CAG CGA GGC* TTTTTTTTTTTTTTTTTTTTT-Cy3 + Stem
S1	AGGGT
S2	GGGTTAGGG

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
