# Peer review of "Impact of Small Molecules on Intermolecular G-Quadruplex Formation"

_molecules, 2019, doi:10.3390/molecules24081570_

Round 1

Reviewer 1 Report

The manuscript is of high importance but it requires some more effort:

1.       The abstract is too general – does not give the idea of the results

2.       There is no information concerning statistics

3.       There is no clear message – what in fact are the sensors supposed to detect/monitor?

4.       There are no in vitro/cytotoxicity results that would indicate the potential of the sensors

Author Response

We have uploaded a Response to Reviewer-1.pdf file as our response.

Reviewer 2 Report

In this study, Gyawali et al. studied the impact of a few small molecules on the kinetics of intramolecular G-quadruplex formation, using FRET technique. The topic is suitable for the journal scope and current special issue, and of general scientific interests. However, the idea and technique are not novel. The G-quadruplex sensor they presented in the study was sensitive in some cases. Yet, I am afraid the data the authors presented are not strong enough to be published in the journal, which is now having a stronger impact.

1. FRET was the only technique used in this paper. In order to confirm the FRET was indeed the reflection of G4 formation (or not), the authors should confirm the structures using other methods such as CD spectrum and native gel electrophoresis.

2. In the case of 3+1 quadruplex, in Figure 1B and 1E, the authors should test plus and minus SM in the same concentration (ie 0.2, 0.5 uM for No SM), so that the readers will have a better appreciation of the SM effects.

3. For the three SMs tested, the authors were only able to successfully get good FRET for L2H2. The authors claimed that there was an acceptor quenching problem. If that was the case, the authors should consider a different pair of fluorophores for the FRET experiments, which will strengthen the data.

4. The authors should repeat the experiments and show the data with error bars, ie in figures 1D, 1G, 3DGJM.

5. Only one sequence was used for each experiment. To reveal a general mechanism and showcase the wide application of this method, the authors have to test other sequences, such as promoter G4 sequences. In fact, these sequences may be more biologically relevant in the case of intramolecular G4.

Author Response

We have uploaded a Response to Reviewer-2.pdf file as our response.

Reviewer 3 Report

FRET-based study that includes a number of conjectures as to the folds and architectures adopted without evidence. FRET alone is a very poor technique for characterization of architecture. Indeed, the topologies described are simply speculation. Appropriate techniques for characterization of the suggested topologies have to be employed. Moreover, the authors should specifically query stoichiometry; since it is a significant part of the conclusions. Secondly, in this particular case of experimental design that includes FRET signal molecules are a significant part of the system, influencing it in many different ways.

Author Response

We have uploaded a Response to Reviewer-3.pdf file as our response.

Reviewer 4 Report

The manuscript entitled “Impact of Small Molecules on Intermolecular G-2 quadruplex Formation” by Balci et al. is well understandable. The main goal is clearly explained.  As a model authors used a system of small ligands pyridostatin, phen-DC3, oxazole telomestatin 32 derivatives and G-quadruplex derived from human telomeric repeat. The drawing of these representative ligands would also be appropriate to be included in the manuscript.

In principle the results of two related arrangements are presented: pd-S3 and pd-S2. However, the results of negative control are not presented in the manuscript.

In addition, I strongly recommend to also include the following results in the manuscript or at least to discuss about the systems: pd-S1 and pd-S4. The systems pd-S1 and pd-S4 are TGG CGA CGG CAG CGA GGC A (GGG TTA)4G-Cy3 + Stem and TGG CGA CGG CAG CGA GGC A GGG TTA G-Cy3 + Stem, respectively.  

What does abbreviation of TMAA mean?

Author Response

We have uploaded a Response to Reviewer-4.pdf file as our response.

Round 2

Reviewer 1 Report

I found all my concerns addressed.

Author Response

We thank the reviewer for her/his positive evaluation.

Reviewer 3 Report

As with the previous version of the manuscript: there is no evidence that there is formation of quadruplexes. What if it turns out that the architectures being formed are indeed higher order but not quadruplexes?

The authors state: "However, we do not actually make any claims about the structural details of the i-GQ structure, e.g. folding conformations."

There is no need to specify which topologies of quadruplexes are being addressed- this would indeed benefit the paper; but the authors do not show any evidence that these are indeed quadruplexes.

The authors also state: "Our control measurement for this conclusion is that the structural compaction we observe takes place only in KCl but not in LiCl or TMAA, which is consistent with i-GQ formation." I disagree- this is simply not a control indicating formation of quadruplexes. It is indeed consistent with compaction- but can be through formation of any higher order architecture; not only quadruplexes. The authors have to prove that quadruplexes are formed- and not any other architecture.

I would still like to see the paper published, since the authors have made an effort to improve other parts of it. However, they have to provide valid evidence for quadruplex formation.

Author Response

We thank the reviewer for her/his recognition of the improvement in the manuscript, which would not have been possible without the time and effort of the reviewers and their constructive comments.

However, the reviewer is still not convinced that the compaction we observe in KCl but not in LiCl or TMAA and the new structures that form in PAGE measurements are convincing enough to demonstrate i-GQ formation. The reviewer considers formation of another type of higher order structure to be possible and expects to see a proof of i-GQ formation. We agree that we do not have a proof of i-GQ formation in a strict analytical sense. However, we disagree with the following statement of the reviewer :”However, they have to provide valid evidence for quadruplex formation.”. Even though our data does not prove i-GQ formation, we believe it provides an evidence that a significant part of the community considers to be valid. All fluorescence-based articles that have been published on GQ structures would have to be classified as invalid if the criterion suggested by the reviewer is applied. In fact, many widely-used and generally-accepted methods (gel electrophoresis, circular dichroism, thermal melting studies, DMS footprinting, optical tweezers, magnetic tweezers…) would need to considered invalid as far as proving i-GQ formation is concerned (or intramolecular GQ formation for that matter). In all of these methods, there is a possibility that another type of higher order structure has formed and gives rise to the signal that is interpreted as an i-GQ. To illustrate, observing a peak at 260 nm and trough at 240 nm in a CD spectrum in KCl but not in LiCl is widely accepted as evidence (not proof) for parallel-GQ formation. This is not a proof (in a strict analytical sense) of parallel-GQ formation, but is considered a valid evidence and a reasonable interpretation. We believe, single molecule FRET is not less valid than any of these methods.

Since the reviewer has not explicitly stated it, we can only infer that s/he might wish to see X-ray crystallography or NMR type structure as a proof of i-GQ formation. If that is the case, unfortunately, we do not have these capabilities and do not think it would be a reasonable expectation from us. Such a criterion would have probably made a large majority of all the GQ literature invalid as only a small fraction of it use such high resolution methods. Even in studies that use the similar sequences as those used in high-resolution structural studies, one can argue that unless the salt conditions, nucleic acid concentrations (i.e. crowding), pH, flanking sequences, annealing and sample preparation protocols also have to be identical in order to conclude that the same higher order structure forms since we know that all of these factors do influence what type of structures form. Therefore, we believe reasonable interpretations rather than analytical proofs should be expected from studies on these complex macromolecules.

If these arguments are not considered adequate to convince the reviewer, we believe the editor should make a decision about the manuscript based on cumulative of the reviewer reports, as the difference of opinion on the matter might be more fundamental than the particulars of this study.    

Reviewer 4 Report

Authors fulfilled all my requirements and answered my comments sufficiently, therefore I have no additional comments.

Author Response

(The authors gave the same response as above.)
